# Impact of Complex Apoptotic Signaling Pathways on Cancer Cell Sensitivity to Therapy

**DOI:** 10.3390/cancers16050984

**Published:** 2024-02-28

**Authors:** Ryungsa Kim, Takanori Kin, William T. Beck

**Affiliations:** 1Department of Breast Surgery, Hiroshima Mark Clinic, 1-4-3F, 2-Chome Ohte-machi, Naka-ku, Hiroshima 730-0051, Japan; 2Department of Breast and Endocrine Surgery, Osaka University Graduate School of Medicine, Suita, Osaka 565-0871, Japan; ymj5014266@gmail.com; 3Department of Pharmaceutical Sciences, College of Pharmacy, University of Illinois at Chicago, Chicago, IL 60612, USA; wtbeck@uic.edu

**Keywords:** anticancer drug, signaling pathway, cell death, antitumor immunity, cancer cell

## Abstract

**Simple Summary:**

We summarize the current knowledge of the signaling pathways involved in anticancer drug-induced cell death. We discuss the common signaling pathways of apoptotic cell death, antiapoptotic pathways, non-apoptotic cell death mechanisms (autophagic, necrotic, and other), signaling pathways involved in the death of drug-sensitive and -resistant tumor cells (with emphasis on *c-Jun*/activator protein 1 and crosstalk with mitochondrial and endoplasmic reticulum pathways), and therapeutic implications of the modification of signaling pathways leading to cell death (with emphasis on cell death-related gene targeting, interactions of drug resistance factors in drug-resistant cells, and the unfolded protein response pathway). We provide suggestions for the restoration of these altered signaling pathways to potentially restore the drug sensitivity of tumor cells.

**Abstract:**

Anticancer drugs induce apoptotic and non-apoptotic cell death in various cancer types. The signaling pathways for anticancer drug-induced apoptotic cell death have been shown to differ between drug-sensitive and drug-resistant cells. In atypical multidrug-resistant leukemia cells, the *c-Jun*/activator protein 1 (AP-1)/*p53* signaling pathway leading to apoptotic death is altered. Cancer cells treated with anticancer drugs undergo *c-Jun*/AP-1–mediated apoptotic death and are involved in *c-Jun* N-terminal kinase activation and growth arrest- and DNA damage-inducible gene 153 (*Gadd153*)/CCAAT/enhancer-binding protein homologous protein pathway induction, regardless of the *p53* genotype. *Gadd153* induction is associated with mitochondrial membrane permeabilization after anticancer drug treatment and involves a coupled endoplasmic reticulum stress response. The induction of apoptosis by anticancer drugs is mediated by the intrinsic pathway (cytochrome c, Cyt c) and subsequent activation of the caspase cascade via proapoptotic genes (e.g., *Bax* and *Bcl-xS*) and their interactions. Anticancer drug-induced apoptosis involves caspase-dependent and caspase-independent pathways and occurs via intrinsic and extrinsic pathways. The targeting of antiapoptotic genes such as *Bcl-2* enhances anticancer drug efficacy. The modulation of apoptotic signaling by *Bcl-xS* transduction increases the sensitivity of multidrug resistance-related protein-overexpressing epidermoid carcinoma cells to anticancer drugs. The significance of autophagy in cancer therapy remains to be elucidated. In this review, we summarize current knowledge of cancer cell death-related signaling pathways and their alterations during anticancer drug treatment and discuss potential strategies to enhance treatment efficacy.

## 1. Introduction

The hallmarks of cancer cells include persistent growth signaling, growth inhibitor evasion, resistance of cell death, unlimited replication capacity, angiogenesis, genomic instability, unleashed phenotypic plasticity, and the evasion of immune destruction [1]. These features allow cancer cells to continue to proliferate, invade neighboring tissues, and spread to other organs to establish metastatic sites. During these processes, the alteration of signaling pathways in cancer cells affects their sensitivity and resistance to anticancer agents and antitumor immunity, making cancer treatment more difficult and less curative.

Anticancer drug-induced cell death can be classified into at least three forms according to morphological and biological criteria: apoptosis, autophagy, and necrosis [2]. Whereas apoptosis and necrosis are irreversible, autophagy is reversible and can lead to cell death or immune escape. The induction of apoptotic cancer cell death is an important component of the therapeutic effects of anticancer drugs [3], and its attenuation correlates with resistance to these drugs [4]. The molecular mechanisms of the intrinsic and extrinsic signaling pathways mediated by mitochondrial outer membrane permeabilization (MOMP) and death receptors (DRs) such as Fas and DR4/5 have been investigated extensively in research on anticancer drug-induced apoptotic cell death [5,6]. The regulation of these pathways is mediated by proapoptotic and antiapoptotic B-cell lymphoma 2 (*Bcl-2*) family proteins. Their modulation through the activation of proapoptotic proteins such as *Bax* and the inhibition of antiapoptotic proteins such as *Bcl-2* enhances the therapeutic efficacy against cancer cells [7]. Necrosis and autophagy are also involved in the therapeutic effects of anticancer drugs [8], but the interaction and relationship between apoptotic and autophagic cell death (ACD) in this context remain to be elucidated [9].

Cancer cells’ development of multidrug resistance (MDR) is a consideration in successful cancer treatment. MDR is caused by multiple factors, including the overexpression of transmembrane proteins as drug efflux pumps via ATP-binding cassette transporters such as P-glycoprotein and multidrug resistance-related protein (MRP), increased levels of detoxification enzymes such as glutathione S transferase, the alteration of DNA target enzymes such as topoisomerase I/II, and the attenuation of DNA damage responses and apoptotic signaling pathways [10,11]. The examination of changes in the signaling pathways leading to apoptosis in drug-resistant cells provides a deeper understanding of the molecular mechanisms underlying the therapeutic effects of anticancer drugs.

Transcription factors such as activator protein 1 (AP-1) and *p53* play important roles in the signaling pathway leading to apoptotic cell death [12]. Activated *c-Jun* N-terminal kinase (JNK) phosphorylates *c-Jun*, which heterodimerizes with the *c-Fos* family as AP-1 and activates DNA damage-inducible genes [growth arrest- and DNA damage-inducible gene 153 (*Gadd153*)/CCAAT/enhancer-binding protein (C/EBP) homologous protein (CHOP)], *Bak*, *Bim*, and *p53* to promote apoptotic cell death [13]. The attenuation or alteration of signaling pathways involved in transcription factor-mediated apoptotic cell death can lead to drug resistance in cancer cells [14]. The attenuation of the DNA damage response by anticancer drugs leads to a decrease in antitumor immune activity due to a reduction in immunogenic cell death (ICD). The induction of ICD activates the ICD signaling pathway by releasing damage-associated molecular patterns (DAMPs) from dying tumor cells, leading to the activation of tumor-specific immune responses, and providing for the long-term efficacy of anticancer drugs [15,16].

In this review, we summarize the current knowledge of the signaling pathways involved in anticancer drug-induced cell death. We will discuss common signaling pathways of apoptotic cell death and antiapoptotic pathways; non-apoptotic cell death mechanisms; signaling pathways involved in the death of drug-sensitive and -resistant tumor cells (including an emphasis on *c-Jun*/AP-1, apoptotic pathways in breast and gastric cell models that highlight these pathways, and crosstalk with mitochondrial and ER pathways); therapeutic implications of the modification of signaling pathways leading to cell death (cell death-related gene targeting, interactions of drug resistance factors in drug resistant cells, and the UPR pathway); and suggestions as to how restoration of these altered signaling pathways may restore drug sensitivity to the tumor cells. We hope to contribute to the understanding of the molecular mechanisms of cell death and provide insights into overcoming cancers’ resistance to therapy, with the overall goal of improving the efficacy of therapy.

## 2. Common Pathways of Apoptotic Cell Death

### 2.1. Intrinsic (Mitochondrial) Pathway

The signaling pathways induced by anticancer drugs are summarized in Figure 1. Mitochondria play a pivotal role in the regulation of anticancer drug-induced apoptotic cancer cell death. In the intrinsic pathway, increased MOMP leads to the release of molecules such as Cyt c, second mitochondria-derived activator of caspase (Smac)/direct inhibitor of apoptosis binding protein with low isoelectric point (DIABLO), and Omi/HtrA2 from the inner mitochondrial space and the activation of a caspase cascade via the activation of the proapoptotic protein *Bax*/*Bak* [17,18,19,20,21]. Cyt c activates caspase 9/3 via apoptosomes composed of apoptotic peptidase activating factor 1 (Apaf-1) and procaspase 9 in the presence of deoxy-ATP or ATP [22]. Smac/DIABLO and Omi/HtrA2 activate the caspase cascade by inhibiting the inhibitor of apoptosis protein (IAP), leading to apoptotic cell death [18,19,20]. *p53* regulates the transcription of downstream proapoptotic target genes such as *Bax*, *Noxa*, *Puma*, and Fas and binds to antiapoptotic proteins such as *Bcl-2* and *Bcl-xL* to increase *Bcl-2* homology domain 3 (BH3)-only proteins such as *Bid* and *Bim,* thereby regulating the *Bax*/*Bad*-mediated apoptotic cell death pathway [23]. *Bcl-xS* inhibits *Bcl-xL*, resulting in the activation of the *Bax*/*Bak*-mediated pathway [24].

JNK is required for the release of Cyt c from mitochondria in apoptotic cell death [25]. Activated JNK promotes the dissociation of *Bax* from this protein by translocation from the cytosol to the mitochondria via phosphorylation of 14-3-3, the cytoplasmic anchor of *Bax* [26]. Mouse embryonic fibroblasts (MEFs) derived from JNK1^−/−^ JNK2^−/−^ mice resist apoptosis in response to diverse genotoxic and cytotoxic stresses, providing evidence that JNK is involved in apoptotic signaling [27,28]. Growth factor-induced antiapoptotic JNK activation is rapid and transient, whereas γ-ray-induced proapoptotic JNK activation is delayed [29]. JNK activation by anticancer drugs is sustained long term in drug-sensitive cells and transient in drug-resistant cells [30]. The transfection of a dominant-negative JNK allele inhibited JNK activity and blocked anticancer drug-induced apoptosis in drug-sensitive cells [30].

JNK contributes to the phosphorylation of *p53* family proteins in the apoptosis signaling pathway [31], which likely involves the *p53*-mediated upregulation of proapoptotic genes such as *Bax* and *Puma* [32]. JNK activation induced by DNA damage also stabilizes and activates *p73*, a member of the *p53* family that induces genes such as *Bax* and *Puma* [33,34]. Cisplatin-induced *p73*-mediated apoptosis requires JNK, which phosphorylates *p73*. Mutations at the *p53* phosphorylation site of JNK inhibit *p73* stabilization by cisplatin and reduce *p73* transcriptional activity, thereby reducing cisplatin-induced apoptosis [33]. JNK induces the expression of proapoptotic genes and decreases the expression of prosurvival genes through multiple transcription factors in a cell type- and stimulus-specific manner.

### 2.2. Extrinsic (Death Receptor-Mediated) Pathway

MOMP induced by *Bax*/*Bak* activation promotes apoptotic cell death, and *Bid* activates *Bax*/*Bak* following activation by caspase-8. Caspase-8, in turn, is activated by the binding of Fas and DR4/5 to the death factor receptor Fas L and tumor necrosis factor-related apoptosis-inducing ligand (TRAIL), which recruits the tumor necrosis factor receptor 1 (TNFR1) death domain protein (TRADD), Fas-related death domain protein (FADD), and procaspase-8 and forms an intracellular death-induced signaling complex (DISC) that activates procaspase-8 [35]. Following TRAIL activation, FADD is recruited after TRADD dissociates and forms complexes with the receptor-interacting protein (RIP) and tumor necrosis factor receptor–associated factor 2 (TRAF2), which mediate cell survival and death through nuclear factor–kappa B (NF-κB) and JNK1, respectively. Caspase-8 proteolytically cleaves *Bid* to form *tBid*, activating the *Bax*/*Bak*-mediated mitochondrial pathway [36]. TNF-α activates caspase-8, which induces JNK to activate *Bid* (*jBid*) through phosphorylation-mediated cleavage and promote the release of Smac and Omi [37]. The inhibition of cellular IAP 1 by Smac and X-linked inhibitor of apoptosis protein (XIAP) by Smac and Omi leads to the activation of the execution factors caspases-3 and -7, leading to apoptosis [38]. Caspase-8 directly activates caspase-3 without amplification of the mitochondrial pathway [39] but also induces lysosome-associated non-apoptotic cancer cell death [40].

JNK is activated by cisplatin treatment, and its sustained activation induces *c-Jun* activation, in turn stimulating Fas L, a downstream gene associated with apoptosis, in sensitive ovarian cancer cells [41]. The inhibition of cisplatin-induced JNK activation prevents this form of apoptosis, and this activation is transient in drug-resistant cells [41]. Stimulation by the selective adenovirus-mediated delivery of mitogen-activated protein kinase kinase 7 (MKK7) or MKK3, upstream activators of JNK, reactivated Fas L expression and increased the susceptibility of resistant cells to apoptotic cell death [41]. TNF-α induced apoptotic cell death in breast cancer cells and mouse fibroblasts via JNK activation, despite the absence of an antiapoptotic inhibitor of the nuclear factor–kappa B (I-κB)/NF-κB pathway, and the inhibition of JNK activation suppressed this process [42]. The duration of JNK activation by anticancer drugs may be important for the induction of apoptotic cell death.

### 2.3. Antiapoptotic Pathway

The release of apoptotic small molecules by MOMP via *Bax*/*Bak* is an essential event in the caspase-dependent and caspase-independent apoptotic pathways and is inhibited by antiapoptotic proteins such as *Bcl-2* and *Bcl-xL* [43]. These proteins inhibit the migration and oligomerization of *Bax* before it is inserted into the mitochondrial outer membrane. *Bcl-xL* inhibits DISC formation and *Bid* activation by caspase-8, suggesting that it regulates not only the mitochondrial pathway but also the upstream receptor-dependent pathway [44]. *Bcl-2* partially inhibits DR-dependent pathways. MOMP proceeds via the loss of mitochondrial membrane potential, which depends on the death trigger and generates reactive oxygen species (ROS) that in turn activate lysosomal enzymes involved in non-apoptotic cell death [45]. *Bcl-2* and *Bcl-xL* prevent the loss of mitochondrial membrane potential and subsequent production of ROS, partially through the antioxidant function of *Bcl-2* [44].

The antiapoptotic phosphoinositide 3-kinase (PI3K)/Akt pathway plays important roles in tumor development and progression [46]. Akt is a serine-threonine kinase composed of three homologous proteins and is activated by hormones and growth factors. It regulates apoptosis-promoting proteins such as *Bax* and *Bad*; *Bax* is phosphorylated to promote heterodimerization with myeloid leukemia cell 1 (*MCL1*) and *Bcl-xL*, which inhibits its translocation to mitochondria, and *Bad* is dephosphorylated to bind and inactivate 14-3-3 [47,48]. Akt also regulates *Bcl-2* expression via cAMP response element binding protein (CREB) and directly inhibits caspase-9 [49,50]. It inhibits *p53* function via the activation of murine double minute 2 (*MDM2*) [51]. I-κB is phosphorylated by Akt and activates NF-κB as an inhibitor of apoptosis [52]. NF-κB activates important antiapoptotic proteins such as *Bcl-xL*, XIAP, and cellular FLICE-inhibitory protein [53]. It inhibits *p27* and induces ABCB1 (*MDR1*) and matrix metalloproteinase-9, which is involved in cancer cell cycle regulation, drug resistance, and metastasis [54,55,56]. NF-κB prevents TNF-α-induced apoptotic cell death by inhibiting the JNK cascade, including the caspase inhibitor XIAP [57], via its antioxidant function, which reduces TNF-α-induced ROS accumulation. The antiapoptotic activity of NF-κB is regulated by the inhibition of ROS accumulation and the regulation of JNK cascade activation [53].

## 3. Non-Apoptotic Cell Death

### 3.1. Autophagic Cell Death (ACD)

Autophagic cell death has recently been classified into three types [58,59]: autophagy-dependent cell death, which is independent of apoptosis and necrosis; autophagy-associated cell death, in which autophagy causes other types of cell death; and autophagy-mediated cell death, which involves standard cell death mechanisms such as autophagy-induced apoptotic cell death. Autophagy is a cytoprotective phenomenon activated by triggers such as nutrient starvation, differentiation, and development of the cell and organism [60]. Tumor cells induce it to escape immunity, promoting tumor progression and metastasis [61]. As an adaptation to metabolic stress, autophagy involves the degradation of various cytoplasmic components in cells and cell organelles for recycling and turnover. It begins with the formation of autophagosomes (double-membrane vesicles) in the cytoplasm; these vesicles engulf organelles and fuse with lysosomes, which break down their contents for recycling into amino acids [62]. ACD occurs when cellular stresses, such as starvation, exceed the basal level of autophagy under normal physiological conditions. Autophagosome formation is mediated by a series of autophagy-related genes (ATGs) [62]. Among mammalian ATG orthologs, *Beclin-1* plays an important role in the formation of autophagosomes and the progression of autophagy; *Beclin-1^+/–^* mutant mice developed various tumors spontaneously, suggesting that it has a tumor suppression function [63]. In contrast, *Beclin-1^−/−^* embryonic stem cells had a strongly altered autophagic response but normal apoptotic responses to serum removal and UV light [63]. Many *Beclin-1* monoallelic deletions have been observed in breast, ovarian, and prostate cancers; the disruption of both *Beclin-1* alleles results in the suppression of cell proliferation and autophagy in vivo [64]. Given that other tumor suppressors, such as death-associated protein kinase and phosphatase and tensin homolog (PTEN), induce ACD, defects in autophagy are implicated in tumorigenesis [65]. Defects in lysosome-mediated autophagy promote carcinogenesis, and the induction of ACD by γ-irradiation or anticancer drugs is associated with the activation of the lysosomal pathway. Autophagy promotes cancer cell adaptation and survival during tumor growth in the tumor microenvironment, but it proceeds to ACD in certain circumstances [66]. ACD in response to anticancer drugs has been reported, for example, in breast, colon, and ovarian cancer tissues [67,68]. Tamoxifen induces ACD in breast cancer cells in association with Akt downregulation; conversely, autophagy is involved in tamoxifen resistance via the activation of the PI3K/Akt/mammalian target of rapamycin (mTOR) signaling pathway [69]. Histone deacetylase inhibitors induce mitochondria-mediated apoptosis and caspase-independent ACD in multiple human cancer cell types [70]. Autophagy is regulated by the mTOR pathway, which in turn has been shown to be regulated by growth factors via the class I PI3K/Akt signaling pathway and by the downregulation of nutrient transporters upon growth factor withdrawal [71]. The activation of epidermal growth factor by extracellular signal-regulated kinase and Akt blocks ACD through crosstalk between the PI3K/Akt and *Ras*/extracellular signaling pathways [72]. In contrast, PTEN, a negative regulator of autophagy, stimulates ACD in colon cancer cells. Akt is downstream of class I PI3Ks and inhibits autophagy by activating the mTOR kinase [73]. Class I PI3K signaling inhibits autophagy by activating it through growth factor receptors, whereas class III PI3Ks promote autophagy by enhancing the sequestration of cytoplasmic components. The presence of autophagic vacuoles in cancer cells killed by anticancer drugs indicates the occurrence of ACD, but the extent to which ACD is involved in the therapeutic effects of these drugs is unclear. When these drugs induce cancer cell cytotoxicity, autophagy is activated and has a protective effect via the degradation of damaged cells. When the cytoprotective response of autophagic vacuoles exceeds the threshold of cytoprotective agents, cancer cells undergo ACD, in which the Golgi apparatus and ER are degraded before nuclear destruction occurs. The treatment of *Bax*/*Bak*-deficient mice with growth factor abolished ACD in interleukin-3 (IL-3)-dependent *Bax*^−^^/−^/*Bak^−/−^* myeloid cells, but these cells underwent ACD after IL-3 deprivation [74]. This reversal of cell viability with the addition of IL-3 suggests that autophagy is a self-limiting survival strategy, rather than an irreversible death program. In *Bax^−/−^ Bak^−/−^* fibroblasts, however, etoposide (VP-16) treatment induced ACD [75]. Growth factor deprivation and drug treatment activate autophagy in *Bax/Bak*-deficient MEFs, with significant differences in the expression of ATGs such as *Atg5* and *Atg6* between the conditions [76]. Low levels of these genes during autophagy result in survival, and high levels result in cell death. The molecular mechanisms of energy crisis and ACD induced by cytotoxic agents may differ, given that autophagy-dependent cell death may have different molecular mechanisms depending on the situation and cell type, and the elucidation of the regulatory mechanisms occurring in these settings is important.

JNK regulates events at various levels in the nucleus and cytoplasm. In the nucleus, it has been shown to upregulate the expression of several ATGs (e.g., *Atg5*, *Atg7*, light chain 3, and *Beclin-1*) in response to certain pro-death stimuli [77]. JNK activation and JNK-mediated ATG expression are required for ACD activation by caspase-8 inhibitors and TNF-α in various cell types. JNK can also activate *c-Jun*, suggesting that AP-1 is involved in JNK-mediated ATGs upregulation [77]. The antiapoptotic *Bcl-2*/*Bcl-xL* protein complex sequesters *Beclin-1* and inhibits autophagy [78]. When the ACD reaction occurs, activated JNK phosphorylates *Bcl-2*/*Bcl-xL*, releasing *Beclin-1* to promote autophagy [79]. Sustained *Bcl-2* phosphorylation leads to *Beclin-1*–mediated ACD. *Bim* inhibits autophagy by recruiting *Beclin-1* in microtubules, whereas activated JNK phosphorylates *Bim*, releasing *Beclin-1* and inducing autophagy [80]. The extent to which ACD contributes to the therapeutic effects of anticancer drugs, and how it crosstalks with and compensates for other cell death types (e.g., apoptotic cell death), remains unclear.

### 3.2. Necrotic Cell Death (NCD)

Necrosis is unscheduled cell death caused by the loss of ATP or dysfunction of the mitochondrial membrane pumps [81]. It is characterized by organelle swelling and cell membrane rupture and is induced by apoptosis (secondary necrosis) [82]. DRs induce not only apoptosis but also non-apoptotic cell death. TNF and TRAIL induce both ACD and NCD, in the former case in wild-type MEFs when caspases are inhibited [83]. TNF-induced necrosis, of which RIP, FADD, and TRAF2 are important components, is mediated by TNFR1 and suppressed by NF-κB. It increased intracellular ROS levels in wild-type, but not RIP^−/−^, FADD^−/−^, or TRAF2^−/−^, MEFs [83]. Anticancer agents such as ethacrynic acid and cytochalasin B induce the necrosis of cervical cancer cells [84]. DNA-alkylating agents, such as cisplatin and other cytotoxic drugs, induce NCD in the absence of *p53*, *Bax*, and *Bak*, regardless of apoptotic defects [85]. In the solid tumor microenvironment, ATP depletion leads to NCD under hypoxic and anoxic conditions, which in turn promotes tumor growth through inflammatory cytokine production rather than having a therapeutic effect. Hypoxia is a hallmark of solid tumors such as gastrointestinal cancers, and *Bcl-2*/adenovirus E1B 19-kDa interacting protein 3 (*BNIP3*), a proapoptotic protein of the *Bcl-2* family, regulates hypoxia-induced cell death triggered by hypoxia-inducible factor 1 (HIF1) [86]. *BNIP3* induces the reduction in the mitochondrial membrane potential and NCD without releasing Cyt c [87]. However, given that the silencing and downregulation of the *BNIP3* gene has been observed in gastric and colorectal cancers, as 5′ CpG island DNA methylation occurs frequently, the inactivation of *BNIP3* is considered to play an important role in gastrointestinal cancer progression [88]. Hypoxia induces the expression of antiapoptotic proteins such as IAP-2 and downregulates the expression of proapoptotic protein *Bax* [86]. In hematopoietic tumors such as acute lymphoblastic leukemia, acute myelogenous leukemia, and multiple myeloma, *BNIP3* expression is suppressed by the abnormal methylation of 5′ CpG islands and histone deacetylases [89]. To escape hypoxia-induced cell death, cancer cells with more aggressive phenotypes that are resistant to anticancer drugs through the inhibition of proapoptotic proteins are selected.

### 3.3. Other Non-Apoptotic Cell Death

Several other forms of non-apoptotic cell death, including ferroptosis, pyroptosis, and necroptosis, are of interest in the context of the eradication of drug-resistant tumor cells. As cancer cells can resist innately programmed and drug-induced apoptosis, effective cancer treatment also requires the identification and targeting of non-apoptotic cell death, although the degree to which these mechanisms act in the anticancer drug resistance of tumor cells is unclear, especially when considered in the context of the mechanisms discussed above. This type of cell death is not mediated by the caspase cascade in the absence of chromatin condensation, nuclear fragmentation, and membrane exudation, which are typical morphological features of apoptosis. Ferroptosis is characterized by the accumulation of iron ions and lipid peroxidation by ROS, and it has morphological and biochemical features that differ from those of apoptosis and autophagy, including mitochondrial contraction, increased membrane density, and the reduction or loss of mitochondrial cristae [90,91]. In certain cancer cell types, ferroptosis-inducing agents reverse drug resistance [92]. The targeting of ferroptosis in cancer cells can reverse resistance to immune checkpoint inhibitors (ICIs), and the interferon-γ signaling pathway is critical for the modulation of treatment outcomes in this context [93]. Necroptosis is characterized by the resistance of apoptotic cell death and involves DRs and RIP kinases; it leads to the rupture of the cell membrane and activation of the immune response [94]. The molecular mechanism of necroptosis depends on receptor-interacting serine/threonine kinase 1 (RIP1), RIP3, and mixed-lineage kinase domain-like pseudokinases, regardless of the trigger [95]. Pyroptosis is a form of programmed cell death characterized by inflammation and is mediated by the gasdermin family [96]. It occurs when certain inflammasomes promote caspase-1 activation, which leads to gasdermin cleavage and the activation of cytokines such as IL-18 and IL-1β [97]. Pyroptosis induces a strong inflammatory response and tumor regression [98]. Crosstalk occurs among apoptosis, necroptosis, and pyroptosis [99]. Again, as indicated, the role(s) of these other non-apoptotic cell death mechanisms in anticancer drug resistance, especially clinical anticancer drug resistance, is unclear.

## 4. Signaling Pathways Involved in the Death of Drug-Sensitive and -Resistant Cells

### 4.1. c-Jun/AP-1

AP-1 plays important roles in cancer cell proliferation, differentiation, and death, depending on the cell type and trigger [100]. Research on *c-Jun*/AP-1 activation and its association with apoptotic cancer cell death is summarized in Table 1. The first report of increased *c-Jun* expression in human myeloid leukemia cells after anticancer drug treatment was published more than three decades ago; the authors suggested that this increase occurred along the differentiation signaling pathway via a drug-induced transcription mechanism [101]. Subsequent studies showed that anticancer drug (VP-16, camptothecin, cytosine arabinoside, and cisplatin)-induced increases in *c-Jun* expression are associated with apoptotic leukemia cell death [102,103,104,105,106]. Some drugs that increase *c-Jun* expression co-induce *c-Fos* expression to a lesser extent, although this expression is not essential for apoptosis [107]. Increased *c-Jun* expression induced by gemcitabine has been associated with apoptotic pancreatic cancer cell death [108]. The activation of JNK and increased expression of *c-Jun* have been associated with the apoptotic death of human leukemia cells treated with paclitaxel (PTX) and epidermoid carcinoma cells treated with PTX and vinblastine (VBL) [109,110,111,112]. In one study, PTX induced JNK activation and increased *c-Jun* expression, but the activated JNK did not phosphorylate *c-Jun*, suggesting that PTX-induced apoptotic epidermal cancer cell death has an AP-1–independent pathway [112]. In epidermoid carcinoma cells treated with VBL, doxorubicin, and VP-16, JNK activation was associated with apoptotic cell death, but only VBL induced the phosphorylation of *c-Jun* and activation of AP-1 [113]; JNK activation was also associated with apoptotic cell death in head and neck squamous cell carcinoma cells treated with PTX [114].

We demonstrated some time ago that increased *c-Jun* expression and AP-1 activation were associated with apoptotic death after VM-26 treatment in drug-sensitive leukemia cells but were attenuated according to the degree of VM-26 resistance [115]. We further found that the dimerization partner of *c-Jun* in AP-1 activation was *Fra-1* in drug-sensitive cells but *Fra-2* in drug-resistant cells, suggesting that the signaling pathway associated with AP-1 activation is altered, leading to apoptotic death, in the latter [115]. Similarly, AP-1 activation and dimerization with *c-Jun* and *Fra-1* were observed in epidermoid carcinoma cells upon VBL treatment, suggesting that Fas L, TNF-α, *Bak*, insulin-like growth factor binding protein 4, and glutathione s-transferase 3 are target genes [110,111]. In contrast, the dimerization partners of *c-Jun* in AP-1 activation in human B lymphoblasts after PTX treatment were *Jun B* and *Jun D* [109]. Attenuated JNK/*c-Jun*/AP-1 activation during apoptosis was observed in cisplatin-resistant human cervical cancer cells [116]. JNK activation was also attenuated in non-small cell lung cancer cells with acquired resistance to gemcitabine [117]. AP-1 activation has been associated with apoptotic cell death induced by anticancer agents such as docetaxel (in gastric cancer cells) [118] and VBL (in epidermal cancer cells) [110,111]. Treatment with doxorubicin, VP-16, or PTX did not activate AP-1 in epidermoid carcinoma cells, suggesting the existence of AP-1-dependent and -independent pathways after JNK activation [112,113]. AP-1 activated by docetaxel appears to target *Gadd153*/CHOP in gastric cancer cells [119] and that activated by VBL appears to target Fas-L, TNF-α, and *Bak* in epidermoid carcinoma cells [111]. Another study suggested that *Bim* was a target gene for apoptotic death in pancreatic cancer cells treated with gemcitabine [108].

Overall, these data suggest that *c-Jun*-associated AP-1 activation, while prominent, is not universal in anticancer drug-induced apoptotic cell death but varies depending on the triggering drug and cancer cell type. Nevertheless, the activation of AP-1 dimerized with *c-Jun* and *Fra-1* and the subsequent activation of proapoptotic proteins may play an important role in at least some signaling pathways leading to anticancer drug-induced apoptosis. In drug-resistant cells, these signaling pathways are attenuated. Clearly, these signaling pathways provide insights and potential targets in anticancer drug-resistant tumor cells.

### 4.2. Apoptotic Pathways Induced by Anticancer Drugs: Gastric and Breast Cancer Cell Models

The following text is summarized in Figure 2. A gastric cancer cell model of the signaling pathway of anticancer drug-induced apoptotic cell death suggests that increased AP-1 activity plays an important role and is associated with *Gadd153* induction, regardless of the *p53* genotype [118]. This finding is consistent with the increased sensitivity of gastric cancer cells to various anticancer drugs upon *Gadd153* introduction, in association with apoptotic cell death and with no alteration of the effects of other drug sensitivity-related factors, such as drug-targeting enzymes and efflux pumps [119]. Another anticancer drug-induced signaling pathway involved in AP-1 activation involves *Bax* induction, which activates the caspase cascade and causes apoptotic cell death. In gastric cancer cell lines, the induction levels of *Bax* and *Bcl-xS* are associated significantly with anticancer drug-induced apoptotic cell death [120]. *Bax* transfection into gastric cancer cells increased their drug sensitivity and co-induced *Bcl-xS* in association with apoptosis [120]. *Bax* co-induction was also observed in gastric cancer cells transfected with *Bcl-xS*, suggesting that the interaction between *Bax* and *Bcl-xS* induces anticancer drug-induced apoptotic cell death [120]. The enhanced drug sensitivity of *Bax*-transfected cells involves the activation of JNK and caspase-3, which leads to apoptotic cell death [121]. Anticancer drug-induced apoptotic death in gastric cancer cells proceeds by caspase-dependent and -independent pathways [122], as protease inhibitors partially block the internucleosomal DNA ladder but not the release of Cyt c. In addition, combination therapy with anticancer drugs and antisense (AS) *Bcl-2* enhanced the therapeutic effect by downregulating *Bcl-2* and upregulating *Bax*, activating a caspase cascade that led to apoptotic cell death [123].

In breast cancer cells, anticancer drug treatment activates the extrinsic pathway via DR4/5 and Fas-mediated caspase-8, which, in combination with the activation of the intrinsic pathway by *Bax* induction, Cyt c release, and caspase cascade, leads to apoptosis [124]. In drug-resistant breast cancer cells, the activation of the extrinsic pathway by caspase-8 via DR4/5 and Fas was inhibited and the activation of the intrinsic pathway by *Bax*, Cyt c release, and the caspase cascade was induced, suggesting that resistance to apoptotic signaling pathways is regulated independently [124]. Combination therapy with anticancer drugs and AS *Bcl-2* enhanced therapeutic efficacy through the downregulation of *Bcl-2* and pAkt and upregulation of *Bax*, which activated the caspase cascade leading to apoptotic cell death [125,126].

### 4.3. Crosstalk with Mitochondria and ER Pathways and the Role of GADD153

Anticancer agents induce ER stress, to which cancer cells respond via two pathways leading ultimately to survival or death. In the case of survival, cancer cells contribute to the acquisition of anticancer drug resistance [127]. Solid tumor cells are exposed to ER stress due to hypoxia and glucose starvation in the tumor microenvironment, which activates the unfolded protein response (UPR) [128]. This response is mediated by three sensors in the membrane that detach from the sensor inhibitor chaperone protein, glucose-regulated protein 78 (GRP78), upon ER stress: inositol-requiring 1 (IRE1), double-stranded RNA-activated protein kinase-like ER kinase (PERK), and activating transcription factor 6 (ATF6) [128]. IRE1 are activated by ER inhibitors such as tunicamycin and thapsigargin and lead to apoptotic cell death when ER damage exceeds a certain threshold. The IRE1 signaling pathway mediates the activation of caspase-12, which is synthesized as an inactive proenzyme and activated only in response to ER stress, via IRE1’s recruitment of TRAF2 to interact with it, forming the IRE1/TRAF2/caspase-12 complex, which in turn is linked to the activation of caspases 9 and 3 in the mitochondrial signaling pathway [129]. JNK is activated following this TRAF2 recruitment and phosphorylates and inactivates *Bcl-2*, inducing apoptotic cell death. JNK induces and phosphorylates *Bim* through IRE1 activation in ER stress-induced apoptosis [130]. The activation of caspase-9 by ER stress occurs without the release of Cyt c and Apaf-1, suggesting that caspase-12 is the direct trigger of caspase-9 activation leading to apoptotic cell death [131]. However, other studies have revealed a link between mitochondria and ER-induced cell death, suggesting that caspase-12 accumulates apoptosis-promoting small molecules such as Smac/DIABLO and Omi/HtrA2 in the cytoplasm and plays important roles in caspase activation and the countering of IAPs [132].

Unresolved excessive anticancer drug-induced ER stress causes structural changes in the ER membrane via *Bax* and *Bak*, allowing Ca^2+^ to migrate into the cytosol, increasing the cytosolic Ca^2+^ level, and activating calpain, a Ca^2+^-dependent cysteine protease [133]. Subsequently, calpain cleaves procaspase-12 and activates the caspase cascade. Increased cytosolic Ca^2+^ induces MOMP, resulting in the release of Cyt c, which in turn activates apoptosomes via Apaf-1, causing apoptosis. Caspase-12 is the only caspase that has been associated with ER-mediated cell death. Upon activation, it is released into the cytoplasm and cleaves procaspase-9. Despite the absence of functional caspase-12 in most humans [134], ER stress-mediated apoptosis is common in human neurodegenerative diseases [135].

The transcription factor *GADD153* plays an important role in the regulation of ER stress-induced cell death, as *Gadd153*-deficient cells are resistant to such death [136]. *Gadd153* transcription is increased in response to ER stress, and *Gadd153* overexpression induces cell cycle arrest and apoptotic cell death [137]. ER stress activates PERK/ATF4, IRE1, and ATF6 after dissociation from GRP78. *Gadd153* is induced by the heterodimerization of ATF4 and C/EBP-β and regulates *Bcl-2* family proteins such as *Bcl-2* and *Bax* [138,139]. It suppresses *Bcl-2* expression by dimerizing with CREB, and Akt/CREB dimerization induces *Bcl-2* expression [49,139]. The suppression of *Bcl-2* and induction of proapoptotic proteins by *Gadd153* may increase the susceptibility of cancer cells to the mitochondria-dependent apoptotic pathway. The inhibition of *Bcl-2* by *Gadd153* depletes intracellular glutathione, which generates ROS [139]. ER stress-induced apoptotic cell death via *Gadd153* involves mitochondria-dependent and -independent pathways. *Gadd153* is involved in suppressing neuronal death caused by cerebral ischemia, and the induction of its expression by DNA damage is part of the ER stress response to cell death, but this process depends on the death trigger and cell type [140].

*Gadd153* expression is induced by the activation of JNK/AP-1, which dimerizes with *c-Jun* to promote apoptotic cell death in response to ER stress. JNK/AP-1 activation in response to DNA damage activates *Bax*/*Bak*, which in turn increases MOMP and activates a caspase cascade leading to apoptosis [141]. Docetaxel treatment increases AP-1 binding activity and induces *Gadd153* expression, leading to apoptotic cell death in gastric cancer cells [118], and the induction of *Gadd153* expression sensitizes calls to anticancer agents such as VP-16 and cisplatin, leading to apoptosis via the coactivation of *Bax* and JNK and the downregulation of *Bcl-2* [119]. The induction of *Gadd153*/*c-Jun* expression by anticancer drugs has been associated with enhanced drug sensitivity and apoptotic cell death in vitro and in vivo in a variety of human cancer cells: cisplatin in ovarian [142,143,144], melanoma, and head and neck cancer cells [142,145]; PTX in ovarian cancer cells [142]; VP-16 in leukemia cells [146]; 5-fluorouracil and cisplatin in gastric cancer cells [147]; and doxorubicin in breast cancer cells [148]. The relationship of *Gadd153*/*c-Jun* activation to drug sensitivity is summarized in Table 2.

### 4.4. A Summary Model Featuring the Central Role for JNK/c-Jun/AP-1 Signaling Pathways

The above description of signaling pathways involved in the death of drug-sensitive and -resistant tumor cells is summarized in Figure 3. Central to all is the activation of the JNK/*c-Jun*/AP-1 signaling pathways. DNA and microtubule damage caused by anticancer drugs activates JNK, which in turn activates the *c-Jun*/AP-1 signaling pathway and induces target genes such as *Bax*/*Bak* and Fas L, resulting in apoptotic cell death via the activation of the caspase cascade. JNK phosphorylates *Bcl-2*/*xL* to inactivate its function, promotes the oligomerization of *Bax* in apoptosis, and phosphorylates *Bcl-2*/*Beclin-1* to induce autophagic cell death. It and *p53* activate *Gadd153*, leading to ER-mediated apoptotic cell death. The JNK/*c-Jun*/AP-1 pathway plays multifunctional roles in cell proliferation, differentiation, and death, and is known to be involved in tumor progression and metastasis, depending on the circumstances. It can result in cell survival or death, depending on the target gene activated by the magnitude of the death trigger in the tumor microenvironment.

## 5. Therapeutic Implications of the Modification of Signaling Pathways Leading to Cell Death

### 5.1. Cell Death-Related Gene Targeting

Based on the above considerations, one can conceive of the introduction of apoptotic genes or the inhibition of antiapoptotic genes to promote apoptotic cell death, all to enhance the therapeutic effects of anticancer drugs. Preclinical studies have shown that the transfection of proapoptotic genes such as wild-type *p53*, *Bax*, and *Gadd153* into esophageal and gastric cancer cells increases their sensitivity to anticancer drugs [119,121,149]. However, a major obstacle to the progression to clinical trials is that no safe, specific, and efficacious means of target gene introduction using a virus or non-viral vector has been established. Antisense (AS) *Bcl-2* oligonucleotides and small-molecule BH3 domain mimetics that function as *Bcl-2*/*Bcl-xL* inhibitors have been employed for the inhibition of antiapoptotic genes such as *Bcl-2* in preclinical and clinical trials [150,151]. These have demonstrated some clinical efficacy against hematological malignancies and melanoma, but not against solid tumors such as breast and gastrointestinal cancers [152]. Several clinical trials have failed due to the adverse effects and limited efficacy of these approaches, despite the potential of this technology in cancer treatment. The targeting of apoptosis-related genes has so far generally failed to advance to the clinical setting as expected and remains aspirational but encouraged by the recent FDA approval of the *Bcl-2*-specific inhibitor venetoclax for the treatment of chronic lymphatic leukemia (CLL) and acute myeloid leukemia (AML) [153]. Venetoclax was developed as a BH3-mimetric drug, a highly selective *Bcl-2* inhibitor that induces apoptosis in *Bcl*-2-expressing hematological malignancies [154]. Venetoclax was highly effective in patients with relapsed or refractory CLL, and the combination of venetoclax with the anti-CD20 monoclonal antibody, rituximab, resulted in complete remissions in 51% of CLL patients and disease-free survival for up to 2 years after completion of therapy [155]. A phase II clinical trial in high-risk relapsed/refractory AML patients treated with venetoclax showed complete response/complete response with incomplete blood recovery in 19% of patients [156]. The clinical benefits of venetoclax are promising not only when used as a monotherapy but also in combination with anticancer drugs as standard therapy. On the other hand, navitoclax, the first BH3-mimetic drug to inhibit *Bcl-2*, *Bcl-xL*, and *Bcl-W,* showed clinical efficacy in CLL patients in clinical trials, but because platelet survival is dependent on *Bcl-xL,* dose-limiting thrombocytopenia has hampered drug development in the clinical setting [157]. Thus, currently, the clinical use of *Bcl-xL* specific inhibitors is not approved due to their on-targeted toxicity to platelets. Several *Mcl-1* specific inhibitors have entered clinical trials, but because *Mcl-1* is critical for cardiomyocyte survival, these trials have been halted due to on-targeted cardiac toxicity [158,159].

Efforts have been made to enhance the therapeutic effects of anticancer drugs by inhibiting autophagy. Although preclinical studies of autophagy-inhibiting quinolones and hydroquinone have shown therapeutic effects, the results of clinical trials conducted with such inhibitors are inconclusive [160]. Findings suggest that the inhibition of autophagy alone does not necessarily enhance the therapeutic effects of inhibitors used alone or in combination with anticancer agents. Whether the induction or inhibition of autophagy enhances the therapeutic effects of anticancer drugs in terms of apoptotic cell death remains a matter of debate, given the abundance of contradictory data obtained with the use of different triggers and cancer cell types [161].

### 5.2. Interplay of Drug Resistance Factors in Resistant Cancer Cells

Efficient means of regulating and manipulating the multiple factors contributing to anticancer drug resistance (i.e., efflux pumps, detoxification and target enzymes, and cell death signaling) to overcome this resistance or restore drug sensitivity remain to be elucidated. Anticancer drug treatments lead to the activation of the PERK/eukaryotic initiation factor 2α (eIF2α)/CHOP signaling pathway in response to ER stress; when signaling does not lead to cell death, however, increased ABCC1/MRP expression has been observed in ER stress-resistant breast cancer cells [162]. Furthermore, the transcription factor C/EBP-β was found to activate ABCB1/P-glycoprotein in breast cancer cells [163]. These findings suggest that survival signaling via ER stress in drug-resistant cells results in crosstalk with the induction of other resistance factors, such as drug efflux pumps. However, the drug resistance of ABCC1/MRP-overexpressing epidermoid carcinoma cells was found to be due to both reduced drug accumulation and *Bcl-xS* expression during apoptotic cell death [164]. Drug sensitivity was partially restored to these cells by *Bcl-xS* transduction, with no effect of the decreased accumulation of vincristine and doxorubicin [164]. These findings suggest that apoptosis-inducing signals and drug efflux pumps interact independently in resistant cells. The interaction of these factors and drug-targeting enzymes needs to be elucidated to determine their effects on each other and the extent of independent functioning. At least one resistance factor can be modulated independently to partially restore the drug sensitivity of resistant cells, but the acquisition of one drug resistance factor may lead to the acquisition of others, resulting in a multifactorial phenotype of resistance. Consistent with this concept, we demonstrated that resistance to apoptotic cell death induced by VM-26 was accompanied by attenuated *c-Jun*/AP-1 activation and a lack of DNA damage-induced responses by constitutively mutant *p53* in the apoptotic signaling pathway in atypical multidrug resistance (at-MDR) leukemia cells with altered topoisomerase IIα [115,165]. Figure 4 is a schematic interpretation of drug resistance factor interactions. Finally, any role(s) of such ABC transporters in the clinical resistance of patients’ tumors to anticancer drugs must be tempered by the fact that there is little compelling evidence that they are responsible for this clinical phenomenon [166].

### 5.3. Targeting the UPR Pathway and Immune Activation

In general, a mild and chronic ER stress-induced UPR leads to tumor progression and drug resistance, a terminal or threshold-exceeding UPR leads to ER-mediated cell death, and a below-threshold UPR does not promote the acquisition of drug resistance [61]. To enhance the therapeutic effect against various types of cancer and non-cancerous diseases, the use of several inhibitors with small molecules targeting ER sensors such as IRE1, PERK, and ATF6 (in combination with anticancer drugs in the case of cancer) has been investigated in preclinical and clinical studies [167,168]. As the UPR signaling pathways mediated by these three sensors are interconnected, the inhibition of one is promptly compensated by another. Thus, the inhibitor target according to the cancer type and the timing of administration to cancer cells, as well as the specificity and toxicity of such approaches, need to be elucidated.

The relationship between antitumor immune activation and the UPR is another important factor. Antitumor immune activation via ICD induced by anticancer drugs such as anthracyclines plays a crucial role in the elimination of cancer cells [169]. ICD is characterized by spatiotemporal surface rearrangements of danger molecules and DAMPs that occur simultaneously with cell death. DAMPs are endogenous molecules that have housekeeping functions in unstressed cells but act as danger signals sensed by the immune system when exposed to cellular stress or injury [170]. They stimulate the adaptive immune system by binding to homologous receptors on innate immune cells such as dendritic cells (DCs), eliciting tumor antigen-specific CD8 T cell-mediated immune responses, thereby eliminating residual cancer cells and establishing immunological memory. The major DAMPs associated with ICD include calreticulin (CRT), ATP, and high mobility group box 1 (HMGB1). The UPR sensor transports key DAMPs such as CRT and ATP to ICD-stressed cancer cell surfaces, alerting the immune system. Three modules are activated simultaneously upon such exposure of CRT in response to the ICD inducers anthracycline and PTX: (i) the ER decay kinase PERK is activated, causing the phosphorylation of the translation initiation factor eIF2α, followed by the partial activation of caspase-8; (ii) caspase-8-mediated cleavage of the ER protein B-cell receptor-associated protein 31 and structural activation of *Bax* and *Bak* occur; (iii) and Golgi passage of CRT pools occurs via soluble N-ethylmaleimide-sensitive factor activating protein receptor-dependent extravesicular secretion [171]. ATP is secreted by annexin 1 and lysosome-dependent mechanisms. Although the UPR is mechanistically linked to exposure to danger signals from stress-killed cancer cells, the choice of transport mechanism utilized by ICD-induced factors is determined by intracellular damage and the stress pathways. Nevertheless, ICD improves the low immunogenicity of tumor cells in the tumor microenvironment, and the release of large amounts of DAMPs such as CRT, ATP, and HMGB1 during ICD induction activates the ICD signaling pathway, in turn promoting DC maturation and activating cytotoxic T lymphocytes (CTLs); thus, antitumor effects are enhanced. ICD occurs in association with autophagy, ferroptosis, pyroptosis, and necroptosis and promotes antitumor immunity [100]. The UPR and immune activation described above are summarized in Figure 5.

### 5.4. Enhancement of Drug Sensitivity via Molecular Therapies Targeting Tumor Growth and Antitumor Immunity

Drug sensitivity is conferred by multiple factors with altered signaling pathways leading to the death of drug-resistant cells. Drug efficacy can be enhanced with therapies targeting molecules involved in tumor growth. Several molecular therapies targeting human epidermal growth factor receptor 2 (HER2) and cyclin-dependent kinases 4 and 6 (CDK4/6) have improved the efficacy of breast cancer treatment and the breast cancer survival rate [172,173]. The clinical indication for the anti-HER2 antibody trastuzumab has been expanded to gastric cancer [174]. A next-generation antibody–drug conjugate (ADC) in which trastuzumab is combined with the topoisomerase I inhibitor deruxtecan (T-DXd) has demonstrated remarkable therapeutic efficacy against HER2-positive advanced metastatic breast cancer that is refractory to conventional HER2-targeting drugs [175]. This therapeutic effect has also been observed in patients with metastatic breast cancer with low or no HER2 expression [176,177], possibly because T-DXd, a topoisomerase I inhibitor, is not cross-resistant to previously administered drugs such as anthracyclines, taxanes, and HER2-targeting agents. When trastuzumab binds to HER2-expressing tumor cells, T-DXd is internalized and the payload is released into cancer cells, penetrating neighboring cancer cells that do not express HER2 through a bystander effect, while circulating HER2-expressing tumor cells also release deruxtecan and remained stable in the periphery [178,179]. This process can cooperate with the activation of natural killer (NK) cells through antibody-dependent cellular cytotoxicity interactions, thereby exerting a therapeutic effect at the tumor site. The nature of interaction between tumor and peripheral sites in the therapeutic response of HER2-expressing tumor cells is unknown. The clinical application of T-DXd has been extended to the standard treatment of advanced metastatic gastric cancer and HER2-expressing/mutated non-small cell lung cancer in the United States [180,181]. Clinical trials are also being conducted to evaluate the therapeutic efficacy of T-DXd against other advanced solid tumors, including ovarian, pancreatic, cholangiocarcinoma, and bladder cancers. Furthermore, the HER3-targeting patritumab–deruxtecan (HER3-DXd) conjugate has shown beneficial activity against hormone receptor (HR)-positive/HER2-negative breast cancer [182]. Another molecularly targeted ADC is sacituzumab govitecan, which consists of the anti-trophoblast antigen 2 (Trop-2) antibody sacituzumab and the topoisomerase I inhibitor SN-38 [183]. Trop-2 is a cell surface protein involved in tumor progression and metastasis that is expressed strongly in, for example, breast and lung cancer cells [181]. Treatment with sacituzumab govitecan was shown to improve the survival of patients with metastatic triple-negative breast cancer compared with physicians’ choice of chemotherapy [184]; a similar survival benefit was observed for patients with metastatic HR-positive/HER2-negative breast cancer, regardless of the Trop-2 expression level [185].

The use of the CDK4/6 inhibitors palbociclib [186], ribociclib [187], and abemaciclib [188] as first- or second-line therapy confers survival benefits for patients with HR-positive/HER2-negative advanced breast cancer. The postoperative adjuvant use of abemaciclib in combination with endocrine therapy (ET) has been shown to benefit survival, regardless of neoadjuvant chemotherapy, in patients with node-positive high-risk breast cancer [189]. Postoperative adjuvant ribociclib treatment for HR-positive/HER2-negative early breast cancer is currently being evaluated in a phase III trial. CDK4/6 inhibitors restore the efficacy of ET, thereby enhancing therapeutic efficacy against HR-positive/HER2-negative breast cancer. The modulation of antitumor immunity via the abemaciclib-induced activation of CTLs may also contribute to the eradication of residual tumor cells after surgical treatment [190].

Cancer stem cells (CSCs) have been hypothesized to regulate tumor growth hierarchically and resist anticancer drugs in the tumor microenvironment [191]. As drug-resistant cells and CSCs are highly mutated and dynamically regulated by tumor microenvironmental factors, especially epigenetic and genetic changes, the characteristics of their resistance factors may overlap. CSCs play important roles in cancer cell survival and tumor growth by being maintained as drug-resistant cells or dormant cells [192]. Regardless of the mechanism by which they arise, their successful elimination by drug molecular targeting and antitumor immune activation can improve therapeutic efficacy and potentially cure cancer [193]. CSCs have several cancer type-specific cell differentiation markers; they form heterogeneous populations with ambiguous, dynamic biology in the contexts of tumor progression and the microenvironment [192]. Specific targeted therapies against proliferative signaling pathways such as Wnt/β-catenin, Hedgehog, and Notch do not eliminate CSCs or effectively improve therapeutic efficacy in terms of inhibitor specificity or low toxicity [194].

Whether the HER2-signaling pathway of tumor growth is involved in the initiation and progression to CSCs is unknown; cluster of differentiation (CD)44^high^CD24^low^HER2^low^ breast CSCs respond poorly to trastuzumab alone, but the combination of trastuzumab and pertuzumab is effective against these cells [195]. HER2^low^-expressing tumor cells at peripheral sites may have CSC characteristics. This possibility is consistent with the finding that pathological complete response after neoadjuvant chemotherapy is a prognostic factor for patients with HER2-overexpressing breast cancer, but it is not a surrogate marker for event-free survival or overall survival, which depends on the tumor size and nodal status [196,197]. T-DXd may contribute to the elimination of CD44^high^CD24^low^HER2^low^ breast CSCs and cure breast cancer. In addition, significant immune activation by CDK4/6 inhibitors and other anticancer drugs may play an important role in the elimination of minimal residual disease, including breast CSCs after surgical treatment.

Although ICD induces antitumor immune responses via the activation of T and NK cells, effective immune activation after anticancer therapy is not due simply to immunosuppressive networks in the tumor microenvironment [198]. In an immunosuppressive process, programmed cell death ligand 1 (PD-L1) in tumor cells binds to programmed cell death-1 (PD-1) in CTLs, preventing their attack. Normally, PD-L1 is expressed on antigen-presenting cells (APCs) and interacts with PD-1 on T cells to reduce CTL activity in the immune response; that expressed on tumor cells inhibits the interaction between T cells and APCs, resulting in the evasion of CTL attack [199]. This immune evasion can be blocked by the action of anti-PD-1 or anti-PD-L1 antibodies in the interaction between T and tumor cells. Another immunosuppressant is the cytotoxic T lymphocyte-associated antigen-4 (CTLA-4) molecule, which is recruited from the cytoplasm to the T-cell membrane as a form of immune synapse. CTLA-4 competes with CD28 to bind ligands on APCs but shows greater affinity for B7 family ligands, unbinding CD28 and CD80/86. When CTLA-4 binds to ligands on APCs (CD80–B7-1 and CD86–B7-2), the T-cell response is inhibited, T-cell proliferation is suppressed, and cytokine secretion is reduced, comprising immunosuppression. In addition, high levels of CTLA-4 functionally reprogram helper T cells into regulatory T cells with potent immunosuppressive properties. Anti-CTLA-4 antibodies restore the suppression of the immune response and enhance T-cell activity by inhibiting the competitive response of CTLA-4 [200].

The development of ICIs using anti-PD-1, anti-PD-L1, and anti-CTLA-4 antibodies has improved the therapeutic outcomes of anticancer drug treatments for various advanced metastatic tumors [198]. The use of dual ICIs (containing anti-PD-1 or anti-PD-L1 and anti-CTLA-4 antibodies) with or without other therapies increases drug-related adverse events, but its therapeutic effects against melanoma and non-small cell lung cancer have been studied [201]. The management of such events [202]; the exploration of predictive factors such as the tumor mutational burden, DNA damage response pathways, the tumor immune microenvironment, PD-L1 expression, circulating tumor cells, and microbiota [203]; and the elucidation of ICI resistance mechanisms such as tumor immunogenicity, tumor microenvironmental factors, antigen presentation, and the immune response [204] are necessary to improve treatment efficacy. In addition, several molecular targeting agents have been developed using antibodies, ADCs, and small molecules that inhibit tumor growth signaling. These approaches have the potential to break through partial drug resistance and increase survival rates for patients with cancer [205]. The contents of this session are summarized in Figure 6.

In clinical practice, neoadjuvant therapy, adjuvant therapy, and anticancer drug therapy for metastatic cancers are being delivered according to evidence-based regimens, which provide an opportunity for shared decision-making with cancer patients. However, depending on the patient’s condition and background, it is necessary to select the most appropriate therapy for each individual patient. It is also necessary to consider the molecular mechanisms of cell death, activation of antitumor immune response, and adverse events caused by anticancer drugs, and to use molecular targeted drugs in terms of prolonged survival and curative effect for cancer patients. Regarding this latter point, we hope that this review will help stimulate future therapeutic considerations that will eventually translate to clinical practice.

## 6. Conclusions

Our goal in this review was to summarize the many different factors that contribute to the resistance of tumor cells to therapy, focusing on the central role of JNK/*c-Jun*/AP-1 signaling in this phenomenon, signaling that leads to cell death. The diversity of and variation in the signaling pathways of anticancer drug-induced cell death varies among cancer types. However, the activation of apoptotic and non-apoptotic cell death may be a common overlapping pathway for such death. Because of clinical limitations affecting the modification of cell death signaling pathways via the external delivery of target molecules such as antisense molecules and viruses, new drugs that molecularly target growth factor receptors and immune checkpoints, as well as drugs without cross-resistance, are expected to enhance the efficacy of chemotherapy. Successful initial trials and the introduction of new agents for advanced metastatic cancer will lead to the development of curative adjuvant therapy.

Although the development of anticancer drugs and molecular targeted therapies has improved the survival rates of patients with cancer who undergo adjuvant and neoadjuvant therapies, recurrence still occurs, even after curative resection and adequate chemotherapy, due to the emergence of drug-resistant cancer cells and the survival of cancer stem cells (CSCs). Strategies for complete tumor elimination depend on drug sensitivity and antitumor immunity. Increased drug sensitivity results from the use of molecularly targeted and non-cross-resistant drugs against tumor growth and CSCs, and it is associated with the activation of antitumor immunity via ICD. However, the initial drug sensitivity and antitumor immunity in the tumor microenvironment are likely to be individually determined in patients with cancer. Cancer treatment requires the precise individual identification of target molecules for the enhancement of these features. As part of personalized cancer therapy, genomic analysis may reveal predictive factors and lead to the identification of drugs that will work for individual patients. A new approach to the development of mRNA vaccines for personalized antitumor immune activation, used for the 2019 coronavirus vaccine, is the administration of mRNA with neoantigens for individual cancers. Clinical trials are being conducted to verify the therapeutic efficacy of the use of mRNA vaccines in combination with ICIs or anticancer drugs. The activation of tumor immunity to eliminate circulating tumor cells with molecularly targeted drugs, ADCs, ICIs, vaccines, chimeric antigen receptor-T, and other therapeutic strategies will be the key to breakthroughs in cancer therapy. Overall, we hope that this review has highlighted a number of nodes in tumor cells that may be targets for therapeutic intervention to overcome their resistance to therapy.

## Figures and Tables

**Figure 1 cancers-16-00984-f001:**
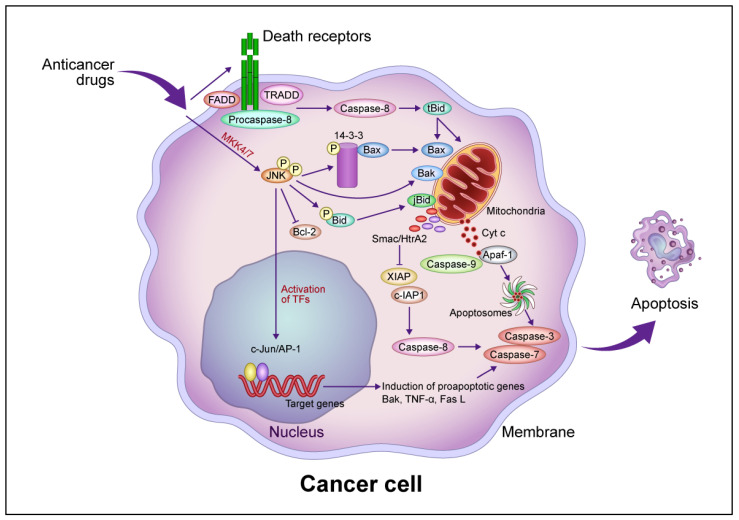
Common intrinsic and extrinsic apoptotic signaling pathways induced by anticancer drugs. In the intrinsic pathway, anticancer drugs activate JNK via mitogen-activated protein kinase kinase (MKK)4/7, which phosphorylates the 14-3-3 protein, dissociates *Bax*, and activates *Bak*. *Bax* and *Bak* translocate to the mitochondrial outer membrane and release Cyt c. Cyt c then forms apoptosomes containing apoptotic protease activating factor 1 (Apaf-1) and procaspase-9, activating the caspase cascade that leads to apoptotic cell death. JNK phosphorylates *Bid* (*jBid*) and migrates through the mitochondria to release second mitochondria-derived activator of caspases (Smac)/high temperature requirement A2 (HtrA2), which inhibits the X-linked inhibitor of apoptosis protein (XIAP) and cellular inhibitor of apoptosis protein (c-IAP), resulting in the activation of caspase-8 and the caspase cascade. JNK also phosphorylates B-cell lymphoma 2 *(Bcl-2*) and inhibits its function. In the extrinsic pathway, anticancer drugs activate death receptors that recruit the Fas-associated death domain (FADD), tumor necrosis factor (TNF) receptor-associated death domain (TRADD), and procaspase-8 to form a death-induced signaling complex, which in turn activates caspase-8. Caspase-8 cleaves, generating truncated *Bid* (*tBid*), which translocates to the mitochondria, releasing proapoptotic proteins such as Cyt c and Smac/HrtA2. JNK activates *c-Jun*/activator protein 1 (AP-1), which induces apoptosis-promoting genes such as *Bak*, TNF-α, and Fas L, in turn activating the caspase cascade leading to apoptotic cell death. Abbreviation: TF, transcription factor. This figure was custom-made by Wiley Editing Services based on our freehand drawing.

**Figure 2 cancers-16-00984-f002:**
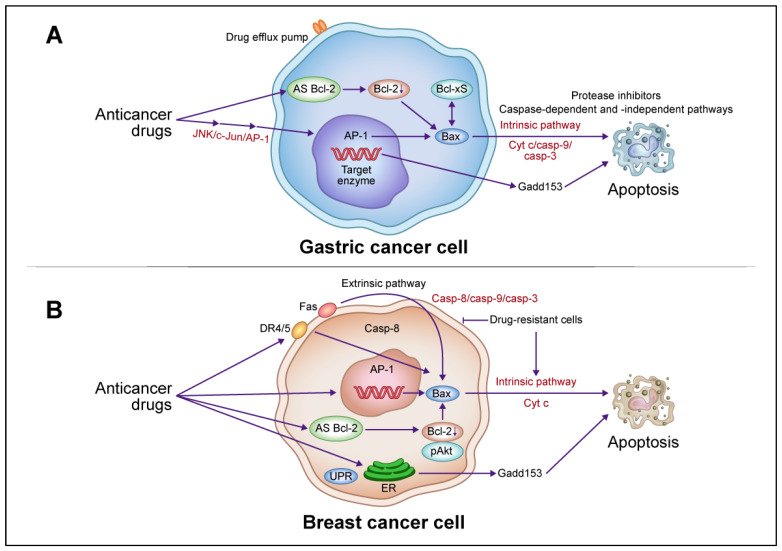
Schematic descriptions of the apoptotic death pathways induced by anticancer drugs in gastric and breast cancer cells. (**A**) In gastric cancer cells, anticancer drugs activate *c-Jun* N-terminal kinase (JNK)/activator protein 1 (AP-1), induce *Bax*, release Cyt c, and activate the caspase cascade leading to apoptotic cell death via an intrinsic pathway. AP-1 activation is involved in growth arrest- and DNA damage-inducible gene 153 (*Gadd153*) induction. Induced *Bax* and B-cell lymphoma (*Bcl*)-*xS* interact to promote this death, which involves caspase-dependent and -independent pathways. Combination therapy with anticancer drugs and antisense (AS) *Bcl-2* enhances the therapeutic effect by downregulating *Bcl-2* and upregulating *Bax*, activating a caspase cascade leading to apoptotic cell death. (**B**) In breast cancer cells, anticancer drugs activate extrinsic pathways via death receptor (DR)4/5 and Fas, leading to the activation of caspase-8 and *Bax*, release of Cyt c, activation of the caspase cascade, and ultimately apoptotic cell death. In drug-resistant cells, anticancer drugs block extrinsic pathways but activate intrinsic pathways involving the induction of *Bax*, Cyt c, and the caspase cascade, leading to apoptosis. Combination therapy with anticancer drugs and AS *Bcl-2* enhances therapeutic efficacy through the downregulation of *Bcl-2* and phosphorylated (p)Akt and upregulation of *Bax*, which activates the caspase cascade and leads to apoptotic cell death. Abbreviations: Casp, caspase; UPR, unfolded protein response; ER endoplasmic reticulum. This figure was custom-made by Wiley Editing Services based on our freehand drawing.

**Figure 3 cancers-16-00984-f003:**
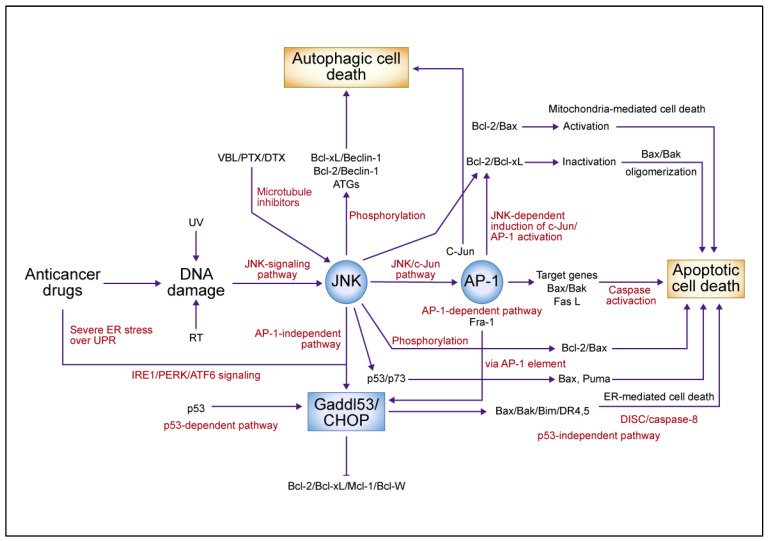
A summary model for the *c-Jun* N-terminal kinase (JNK)/*c-Jun*/activator protein 1 (AP-1)-mediated signaling pathway in anticancer drug-induced cell death. Anticancer drugs that damage DNA and tubulin activate JNK, causing *c-Jun*/AP-1 to dimerize predominantly with *Fra-1* and activate target genes such as *Bax*/*Bak* and Fas L, which activate the caspase cascade in an AP-1-dependent pathway leading to apoptotic cell death. JNK phosphorylates and inactivates antiapoptotic proteins such as B-cell lymphoma (*Bcl*)-*2* and *Bcl-xL*, which in turn activate *Bax* and cause apoptotic cell death. JNK also activates growth arrest- and DNA damage-inducible gene 153 (*Gadd153*), which is induced by AP-1-dependent and -independent pathways in endoplasmic reticulum (ER)-mediated cell death. ER stress and *p53* induction by anticancer drugs activates *Gadd153* and causes ER-mediated cell death. JNK phosphorylates *Beclin-1*, dissociates from *Bcl-2*, and induces autophagic cell death. The JNK-mediated upregulation of autophagy-related genes (ATGs) involves the AP-1 transcription factor complex. In response to DNA damage, JNK mediates apoptosis by phosphorylating *p53*, stabilizing *p73*, dimerizing *p53* and *p73*, and promoting the expression of proapoptotic target genes such as *Bax* and *Puma*. Abbreviations: VBL, vinblastine; PTX, paclitaxel; DTX, docetaxel; UV, ultraviolet; RT, radiation therapy; UPR, unfolded protein response; IRE1, inositol-requiring 1; PERK, protein kinase-like ER kinase; ATF6, activating transcription factor 6; CHOP, CCAAT/enhancer-binding protein homologous protein; DR, death receptor; DISC, death-induced signaling complex; *Mcl-1*, myeloid cell leukemia 1. This figure was custom-made by Wiley Editing Services based on our freehand drawing.

**Figure 4 cancers-16-00984-f004:**
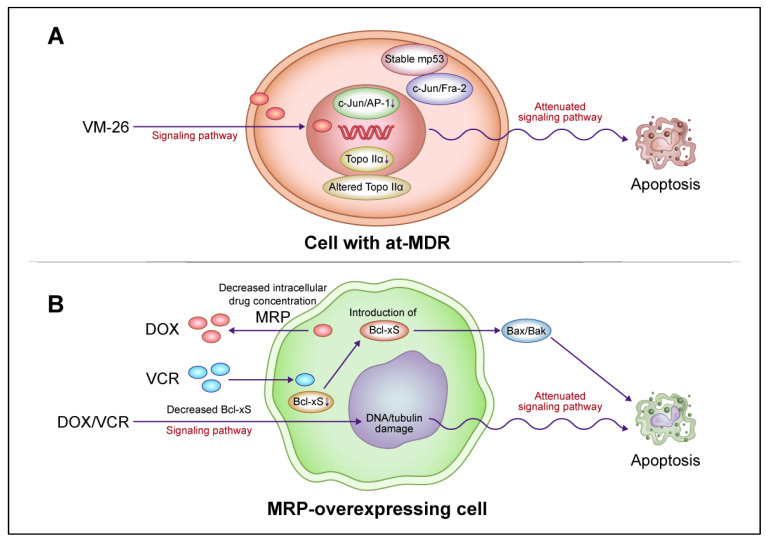
A model for the interaction of drug resistance-related factors in drug-resistant cancer cells. (**A**) Altered signaling pathways for apoptotic cell death mediated by *c-Jun* N-terminal kinase (JNK)/activator protein 1 (AP-1) and mutant *p53* (m*p53*) in lymphoblastic leukemia cells with atypical multidrug resistance (at-MDR). at-MDR is defined by the mutation and reduced expression of topoisomerase IIα (Topo IIα) without drug efflux pump overexpression. Teniposide (VM-26) treatment suppresses *c-Jun*/AP-1 activation in proportion to the degree of drug resistance. Activated AP-1 consists of a *c-Jun*/*Fra-1* dimer in drug-sensitive cells and *c-Jun*/*Fra-2* and, to a lesser extent, *c-Jun*/*Fra-1* dimers in drug-resistant cells. m*p53* induction is more stable and the DNA damage-induced *p53* response is attenuated after VM-26 treatment in resistant cells compared with those in sensitive cells. In resistant cells, alterations in apoptotic cell death signaling pathways and target enzymes are regulated independently. (**B**) Alteration of apoptotic cell death signaling pathways upon the reduction in the proapoptotic protein B-cell lymphoma (*Bcl*)-*xS* in ABCC1/multidrug resistance protein (MRP)-overexpressing epidermoid carcinoma cells. The introduction of *Bcl-xS* promotes apoptotic cell death and partially restores drug sensitivity in resistant cells without affecting the reduction in drug accumulation. The attenuation of the apoptotic cell death pathway via the reduction in *Bcl-xS* expression and drug accumulation by MRP are regulated independently in resistant cells. Abbreviations: DOX, doxorubicin; VCR, vincristine. This figure was custom-made by Wiley Editing Services based on our freehand drawing.

**Figure 5 cancers-16-00984-f005:**
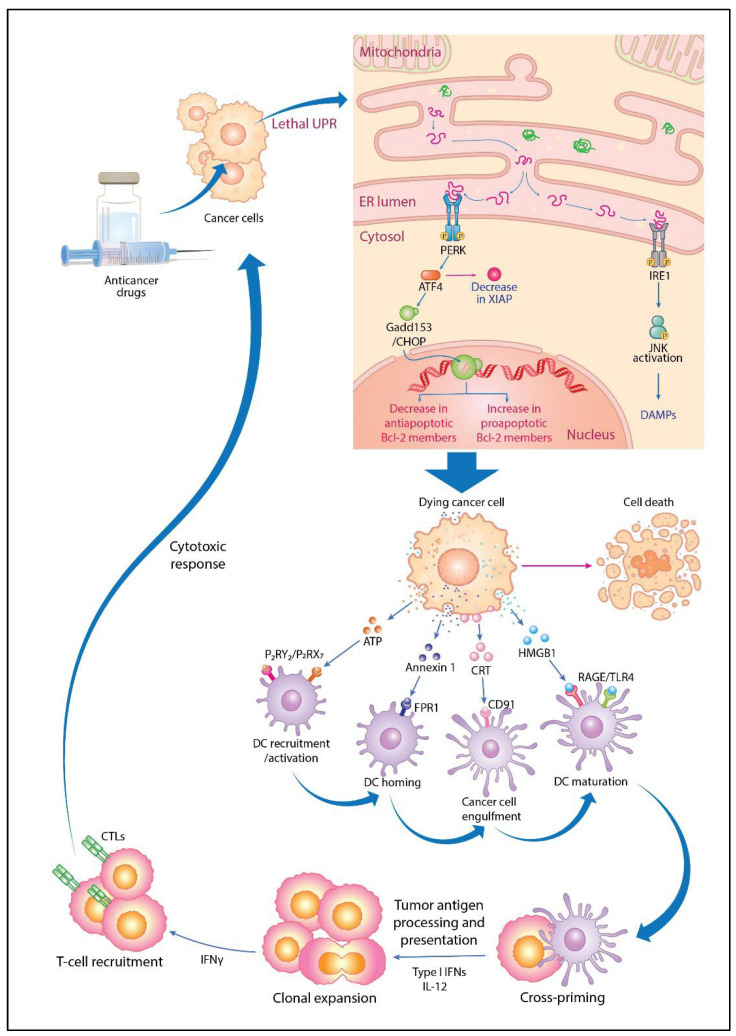
Molecular mechanisms of immunogenic cell death and tumor-specific immune activation via unfolded protein response (UPR) activation by anticancer drugs. When anticancer drugs are administered, cancer cells exceed the UPR threshold to the point of lethality, activating the double-stranded RNA-activated protein kinase-like ER kinase (PERK) and inositol-requiring 1 (IRE1) signaling pathways and activating transcription factor 4 (ATF4) and growth arrest- and DNA damage-inducible gene 153 (*Gadd153*)/CCAAT/enhancer-binding protein homologous protein (CHOP) induction, which in turn downregulates X-linked inhibitor of apoptotic protein (XIAP), decreasing antiapoptotic and increasing proapoptotic B-cell lymphoma 2 (*Bcl-2*) protein expression, leading to cell death. In this process, dying cancer cells externalize calreticulin (CRT) to the membrane surface and release damage-associated molecular patterns (DAMPs) such as ATP, annexin 1, and high mobility group box 1 (HMGB1) for immune activation. Dendritic cells (DCs) are recruited and activated by the secretion of ATP that binds to P_2_RY_2_ and P_2_RX_7_ receptors. They are then homed by the binding of released annexin 1 to formyl peptide receptor 1 (FPR1). CRT stimulates the phagocytosis of dying cancer cells by binding to CD91 receptors on DCs. HMGB1 release stimulates DC recruitment via binding to receptor for advanced glycation end products (RAGE) receptors and induces DC maturation via toll-like receptor 4 (TLR4) signaling. Mature DCs migrate to lymph nodes, where they cross-prime and cause the clonal expansion of T cells, including interleukin 12 (IL-12) and type I interferon (IFN). IFN-γ-producing T cells are recruited and exert cytotoxic responses to eradicate cancer cells. Tumor antigens (TAs) released from dying cells are taken up by immature DCs, which then activate and mature. TAs and tumor-associated antigens (TAAs) are processed in mature DCs and presented to CD8+ T cells as major histocompatibility complex I (MHC I) molecules, generating cytotoxic T lymphocytes (CTLs) for tumor-specific immune responses. This figure was custom-made by Wiley Editing Services based on our freehand drawing.

**Figure 6 cancers-16-00984-f006:**
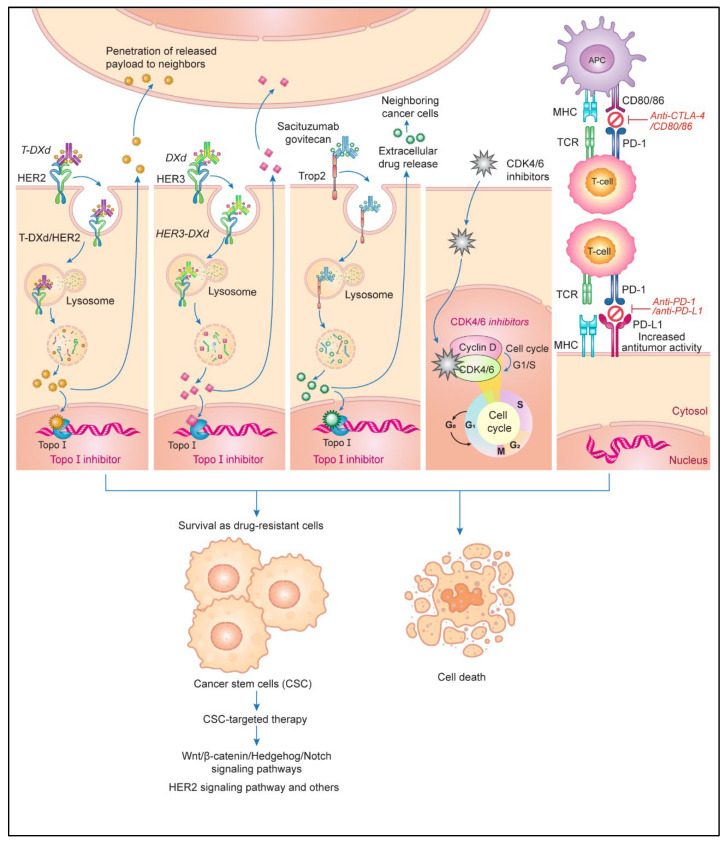
The enhancement of drug sensitivity via molecular therapies targeting tumor growth and antitumor immunity. Human epidermal growth factor receptor (HER) 2- and HER3-targeted antibody–drug conjugate (ADC) products, including topoisomerase I (Topo I) inhibitors such as trastuzumab deluxecan (T-DXd) and patritumab deluxecan (HER3-DXd), have considerable antitumor effects in patients who have undergone HER2-targeted therapy. These ADCs are internalized into cancer cells and release Topo I inhibitors that are not cross-resistant to anthracyclines and taxanes, leading to cell death. The released payload penetrates adjacent cancer cells through a bystander effect. The ADC sacituzumab govitecan targets trophoblast antigen 2 (Trop-2), a surface protein bound to the Topo I inhibitor SN-38, which is released intracellularly and extracellularly. Cell cycle inhibitors targeting cyclin-dependent kinase 4/6 (CDK4/6) in the G1–S phase transition have shown promise when used in combination with endocrine therapy for patients with hormone receptor (HR)-positive HER2-negative breast cancer. The development of cancer stem cells (CSCs) with persisting drug resistance is a major obstacle in chemotherapy. CSC-targeting therapies, which target several signaling pathways involved in cancer survival, may be employed effectively in combination with conventional therapy to eradicate residual tumor cells for the curing of cancer. The combined use of anticancer drugs and immune checkpoint inhibitors, such as anti-programmed cell death 1 (PD-1), anti-programed cell death ligand 1 (PD-L1), and anti-cytotoxic T lymphocyte antigen-4 (CTLA-4) antibodies, enhances therapeutic efficacy by increasing antitumor immune activity. Abbreviations: MHC, major histocompatibility complex; TCR, T-cell receptor; APC, antigen-presenting cell. This figure was custom-made by Wiley Editing Services based on our freehand drawing.

**Table 1 cancers-16-00984-t001:** Associations between increased *c-Jun*/AP-1 activity and anticancer drug-induced apoptosis in human cancer cells.

Anticancer Drug	Cell Line/Origin	JNK/*c-Jun*	AP-1 Activity/Dimer with *c-Jun*	Potential Target Genes	Apoptosis Correlation	Year [Ref.]
VP-16	HL-60 and U-937/ myeloid leukemia cells	ND/mRNA increase	ND	ND	Yes	1991 [102]
Camptothecin	U-937/ myeloid leukemia cells	ND/mRNA increase Co-increase in *c-Fos*	ND	ND	Yes	1991 [103]
Ara-C	U-937/ myeloid leukemia cells	ND/mRNA increase Co-increase in *c-Fos*	ND	ND	Yes	1991 [104]
Cisplatin	HL-60, U-937, and KG-1/ myeloid leukemia cells	ND/mRNA increase Co-increase in *c-Fos*	ND	ND	Yes	1992 [105]
VP-16	K562 and HL-60/ leukemia cells	ND/mRNA increase	ND	ND	Yes	1994 [106]
Gemcitabine	Panc-1 and SW1990/ pancreatic cancer cells	ND/increase	ND	*Bim*	Yes	2015 [108]
Paclitaxel	RPMI-1788/ B lymphoblast leukemia	Activation/increase	Increase/*Jun B*, *Jun D*	ND	Yes	1998 [109]
Vinblastine	KB-3/ epidermoid carcinoma cells	Activation/increase	Increase/*Fra-1*	Fas-L, TNF-α	Yes	2001 [110]
Vinblastine	KB-3/ epidermoid carcinoma cells	Activation/increase	Increase/ND	TNF-α, *Bak*, IGFBP4, GST3	Yes	2001 [111]
Vinblastine Paclitaxel	KB-3/ epidermoid carcinoma cells	Activation/increase	No increase	ND	Yes	2008 [112]
Vinblastine Doxorubicin VP-16	KB-3/ epidermoid carcinoma cells	Activation/ND	Increase by vinblastine but not by doxorubicin and VP-16/ND.	ND	Yes	2003 [113]
Paclitaxel	OEC-M1/ head and neck squamous cell carcinoma cells	Activation/ND	ND	ND	Yes	2021 [114]
VM-26	CEM, CEM VM-1, and CEM VM-1-5/ lymphoblastic leukemia cells	ND/increase	Increase and attenuation in resistant cells/*Fra-1*; *Fra-2* in resistant cells	ND	Yes	1994 [115]
CDDP	Hela and CDDP resistant cells/cervical carcinoma cells	Activation/increase, attenuation in resistant cells	Increase, attenuation in resistant cells/ND	ND	Yes	2004 [116]
Gemcitabine	H1299/non-small cell lung cancer cells	Activation, attenuation in resistant cells/ND	ND	ND	Yes	2005 [117]
Docetaxel	MKN-1, 28, 45, 74, HSC-39, KATO-III, OKAJIMA, and SH 101/gastric cancer cells	ND	Increase/ND	*Gadd153*/CHOP	Yes	1999 [118]

Abbreviations: AP-1 = activator protein 1; JNK = *c-Jun* N-terminal kinase; ref. = reference; VP-16 = etoposide; ND = not done; Ara-C = cytarabine; TNF-α = tumor necrosis factor-α; IGFBP4 = insulin-like growth factor binding protein 4; GST3 = glutathione s-transferase 3; VM-26 = teniposide; CDDP = cisplatin; *Gadd153* = growth arrest- and DNA damage-inducible gene 153; CHOP = CCAAT/enhancer-binding protein homologous protein.

**Table 2 cancers-16-00984-t002:** Associations between *Gadd153*/*c-Jun* increase and anticancer drug sensitivity in human cancer cells.

Anticancer Drug	Cell Line/Origin/Xenograft, Clinical Sample	*Gadd153*/*c-Jun* Expression and Others	Drug Sensitivity Correlation	Apoptosis Correlation	Clinical Response Correlation	Year [Ref.]
CDDP	Ovarian carcinoma 2008 cells/ Melanoma and head and neck xenografts	Increase in *Gadd153* mRNA/ND	Increase in vitro and in vivo	ND	NA	1994 [142]
CDDP, PTX	Ovarian carcinoma 2008 cells	Increase in *Gadd153* mRNA/ND	Increase in vitro	ND	NA	1996 [143]
CDDP	Ovarian carcinoma 2008 cells and resistant subclones	Increase in *Gadd153* mRNA in sensitive cells/no significant difference in maximum expression of *c-Jun* mRNA	Increase in vivo	ND	NA	1997 [144]
VP-16	U937, HL-60/ leukemic cells	Increase in *Gadd153* mRNA	Increase in vitro	Yes	NA	1997 [145]
CDDP	UMSCC10b/ head and neck carcinoma cells/ stage III/IV head and neck cancer	Increase in *Gadd153* mRNA	Increase in vitro and in vivo	ND	Yes	1999 [146]
5-FU, CDDP	TMK-1, MKN-45, 74/gastric cancer cells/stage IIIB/IV advanced gastric cancer	Increase in *Gadd153* and *c-Jun* mRNA	Increase in vitro and in vivo	ND	Yes	2007 [147]
DOX, tunicamycin	4T1 mouse and MDA-MB-468/triple-negative breast cancer cells	Increase in *Gadd153*/CHOP protein associated with GRP78	Increase in vitro and in vivo	Yes	NA	2014 [148]

Abbreviation: *Gadd153* = growth arrest- and DNA damage-inducible gene 153; ref. = reference; CDDP = cisplatin; ND = not done; NA = not applicable; PTX = paclitaxel; VP-16 = etoposide; 5-FU = 5-fluorouracil; DOX = doxorubicin; CHOP = CCAAT/enhancer-binding protein homologous protein; GRP78 = glucose-regulated protein 78.

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
