# Peer review of "Impact of Complex Apoptotic Signaling Pathways on Cancer Cell Sensitivity to Therapy"

_cancers, 2024, doi:10.3390/cancers16050984_

Round 1
Reviewer 1 Report
Comments and Suggestions for Authors
The manuscript is well-written and comprehensive, showcasing a unique writing style that encourages analytical and critical thinking. Overall, the manuscript is ready for publication, with the minor suggestion of addressing software tools. However, the author's ambitious attempt to cover apoptosis alongside various signaling pathways is noteworthy. The figures are precise and contribute to understanding, and it would be beneficial for the authors to include details on any software tools used in figure construction for transparency. Nevertheless, considering the overall merit of the manuscript, I strongly encourage its acceptance.
Author Response
Reviewer 1:
The manuscript is well-written and comprehensive, showcasing a unique writing style that encourages analytical and critical thinking. Overall, the manuscript is ready for publication, with the minor suggestion of addressing software tools. However, the author's ambitious attempt to cover apoptosis alongside various signaling pathways is noteworthy. The figures are precise and contribute to understanding, and it would be beneficial for the authors to include details on any software tools used in figure construction for transparency. Nevertheless, considering the overall merit of the manuscript, I strongly encourage its acceptance.
We thank the Reviewer for his/her very kind comments. We have addressed his/her concern about software tools in the relevant Figure legends.
Reviewer 2 Report
Comments and Suggestions for Authors
The review article summarize current understanding of signaling pathways involved in anticancer drug–induced cell death, especially JNK/c-Jun/AP-1 signaling, and their alterations during anticancer drug treatment. Based on these, it also discuss potential strategies to enhance treatment efficacy for cancers. The article is well written, and this reviewer has no concern, and think that it is appropriate for publication in Cancer.
Author Response
Reviewer 2:
The review article summarize current understanding of signaling pathways involved in anticancer drug–induced cell death, especially JNK/c-Jun/AP-1 signaling, and their alterations during anticancer drug treatment. Based on these, it also discuss potential strategies to enhance treatment efficacy for cancers. The article is well written, and this reviewer has no concern, and think that it is appropriate for publication in Cancer.
We very much appreciate the Reviewer’s very kind comments. Thank you.
Reviewer 3 Report
Comments and Suggestions for Authors
In this manuscript, Kim and colleagues provide a comprehensive overview of the current knowledge regarding the mechanisms of cell death in cancer cells undergoing pharmacological therapy. The summary begins by introducing readers to the primary types of cell death, as well as the key mechanisms of pharmacological resistance. Subsequently, the authors meticulously define the apoptotic pathways (intrinsic and extrinsic), with a specific focus on the signaling pathways regulated by JNK.
Following this, the authors scrutinize the influence and regulation of JNK, c-jun, and AP-1 in the regulation of autophagic cell death and other non-apoptotic cell death types. Overall, the work is well-structured and highly relevant for a specific niche of individuals working in the field of cellular signaling. The manuscript is appropriately organized and logically presented, with current and pertinent references. The English used in the abstract is suitable and devoid of major errors. No instances of plagiarism were detected. The strengths of the work lie in its focused approach to the regulation of cell death pathways governed by c-Jun/JNK. Additionally, the manuscript includes multiple original and visually appealing images, as well as highly useful tables.
The main drawbacks include the fact that many of the discussed topics, such as cell death mechanisms, have been extensively explored in other reviews, and some of the data may appear repetitive. As minor comments, the following observations are noted:
14 nonapoptotic > non-apoptotic (IDEM 21 and 12 more…)
Gene names must be italicized
107 Cyt c must be stated the first time of appearance (L30)
107 please specify what does low pl mean
120 JNK acronym was previously stated
123 IDEM (Cyt c)
2.3 the title is confusing as the authors discuss not a single antiapoptotic pathway, but multiple pathways rescuing the cells from apoptosis. Please rephrase the tile. It is somehow confusing because there is not a defined and well-known antiapoptotic pathway, but multiple signaling contributing to limit apoptosis induction or development.
Author Response
Reviewer 3:
In this manuscript, Kim and colleagues provide a comprehensive overview of the current knowledge regarding the mechanisms of cell death in cancer cells undergoing pharmacological therapy. The summary begins by introducing readers to the primary types of cell death, as well as the key mechanisms of pharmacological resistance. Subsequently, the authors meticulously define the apoptotic pathways (intrinsic and extrinsic), with a specific focus on the signaling pathways regulated by JNK.
Following this, the authors scrutinize the influence and regulation of JNK, c-jun, and AP-1 in the regulation of autophagic cell death and other non-apoptotic cell death types. Overall, the work is well-structured and highly relevant for a specific niche of individuals working in the field of cellular signaling. The manuscript is appropriately organized and logically presented, with current and pertinent references. The English used in the abstract is suitable and devoid of major errors. No instances of plagiarism were detected. The strengths of the work lie in its focused approach to the regulation of cell death pathways governed by c-Jun/JNK. Additionally, the manuscript includes multiple original and visually appealing images, as well as highly useful tables.
We thank the Reviewer for his/her kind comments and for seeing the strengths of our review in our focus on JNK/c-Jun/AP-1 in the regulation of cell death pathways as well as for the clarity or our Figures and Tables.
The main drawbacks include the fact that many of the discussed topics, such as cell death mechanisms, have been extensively explored in other reviews, and some of the data may appear repetitive.
We agree with the Reviewer that cell death mechanisms have been examined extensively in other reviews, but, as the Reviewer notes, the strength of this review resides in the fact that the focus here is the comprehensive analysis of cell death pathways regulated and governed by JNK/c-Jun/AP-1. Also, Reviewers 1 and 2, in their overall support of our manuscript, would appear to agree with our approach.
Last, it is not clear to us that some of the data are repetitive; if so, it is deliberate, to make a point in the section. Further, without specific guidance from the Reviewer as to which data he/she is referring, we respectfully will leave the data as is in the manuscript.
Overall, we thank the Reviewer again for his/her very careful review of our manuscript and for his/her very kind comments.
As minor comments, the following observations are noted:
14 nonapoptotic > non-apoptotic (IDEM 21 and 12 more…)
Gene names must be italicized
The term nonapoptotic was modified to non-apoptotic and the gene names were italicized.
107 Cyt c must be stated the first time of appearance (L30)
The abbreviation of Cyc c was revised.
107 please specify what does low pl mean
The low pl was specified (L107).
120 JNK acronym was previously stated
The JNK acronym was used.
123 IDEM (Cyt c)
Cyt c was used (L122)
2.3 the title is confusing as the authors discuss not a single antiapoptotic pathway, but multiple pathways rescuing the cells from apoptosis. Please rephrase the tile. It is somehow confusing because there is not a defined and well-known antiapoptotic pathway, but multiple signaling contributing to limit apoptosis induction or development.
We thank the Reviewer for this comment. First, we gently note to the Reviewer that we do not discuss ANTIapoptotic pathways, as indicated in the Reviewer’s comments; rather, we discuss complex multiple APOPTOTIC pathways. Second, we do say “,,, apoptotic signaling pathways …” in the title. Accordingly, with respect, we think the title actually very well captures the sense of multiple and complex apoptotic signaling pathways, and we would like to leave the title as is.
Reviewer 4 Report
Comments and Suggestions for Authors
The review of apoptotic pathways involved in cancer cell therapy is presented in detail; However, the relevance of this process in the area of cancer is diluted by the lack of a specific definition of those cases in which the regulation/induction of apoptosis is important. Indeed, the implication of apoptosis in cancer therapy remains an area under continued scrutiny. This sentiment is well presented by the authors. In this sense, it is better to focus on leukemias or other cancers in which apoptosis is an issue, for example, and in which the natural regulation (selection) of apoptosis is lost (leukemias).
In my opinion, the review is solid and summarizes the most recent advances and molecular understanding of the pathways involved, but it needs to provide information on clinical trials (past and present) and the conclusions of their failure (or success, e.g. , venetoclax). This should be reflected in a detailed table indicating the objectives, the disease and main conclusions of these trials.
Furthermore, I recommend highlighting the relevance of ferroptosis (now in minor section 3.3), and that it is a topic of intense research in the area.
Author Response
Reviewer 4:
The review of apoptotic pathways involved in cancer cell therapy is presented in detail; However, the relevance of this process in the area of cancer is diluted by the lack of a specific definition of those cases in which the regulation/induction of apoptosis is important. Indeed, the implication of apoptosis in cancer therapy remains an area under continued scrutiny. This sentiment is well presented by the authors. In this sense, it is better to focus on leukemias or other cancers in which apoptosis is an issue, for example, and in which the natural regulation (selection) of apoptosis is lost (leukemias).
We thank the Reviewer for this comment. We believe that we did show examples of the relevance of apoptosis induction in specific tumors throughout the manuscript (eg, see lines 350-409, 410-459, 460-529 etc; Tables 1 and 2). The Reviewer is correct that apoptosis also has relevance in leukemias and the use of venetoclax (see below) to induce apoptosis in these diseases is important, and we have addressed this in a new paragraph in Section 5 (lines 584-593).
In my opinion, the review is solid and summarizes the most recent advances and molecular understanding of the pathways involved, but it needs to provide information on clinical trials (past and present) and the conclusions of their failure (or success, e.g. , venetoclax). This should be reflected in a detailed table indicating the objectives, the disease and main conclusions of these trials.
We thank the Reviewer for these suggestions. In addition to discussing the use of venetoclax in leukemias, we have added a discussion, also in Section 5 (lines 593-599), of some clinical studies in which drugs induced apoptosis in the tumors. We note that studying apoptosis in solid tumors is not easy, as specimens have to be obtained from the tumors, and apoptosis then measured in vitro. We do not think it necessary to detail all these studies in a new Table; rather, we believe that discussion of some of these clinical studies in the new subsection in Section 5 will be sufficient to make the point. We thank the Reviewer again for these suggestions.
Furthermore, I recommend highlighting the relevance of ferroptosis (now in minor section 3.3), and that it is a topic of intense research in the area.
We thank the Reviewer for this suggestion, and we agree that this is an important subject. However, with respect, we believe that we have adequately covered ferroptosis in Section 3.3, and to give it more attention will, in our opinion, detract from the general emphasis of many of the mechanisms discussed that Section, entitled “Other Nonapoptotic Cell Death”. We hope that the Reviewer will agree with this.